

# Assessing factors influencing students' perceptions towards animal species conservation

Heliene Mota Pereira[1,2], Franciany Braga-Pereira[1,3], Luane Maria Melo Azeredo[1,2], Luiz Carlos Serramo Lopez[1] and Rômulo Romeu Nóbrega Alves[4]

[1] Departamento de Sistemática e Ecologia, Universidade Federal da Paraíba, João Pessoa, Paraíba, Brazil
[2] Programa de Pós-Graduação em Etnobiologia e Conservação da Natureza, Universidade Federal Rural de Pernambuco, Recife, Brazil
[3] Rede de Pesquisa para Estudos sobre Diversidade, Conservação e Uso da Fauna na Amazônia (REDEFAUNA), Manaus, Brazil
[4] Departamento de Biologia, Universidade Estadual da Paraíba, Campina Grande, Paraíba, Brazil

Corresponding author
Heliene Mota Pereira,
heliene.mota@ufrpe.br

## ABSTRACT

**Background:** The way humans perceive and interact with non-human animals is particular to each person, from antipathetic interactions evidenced by fear, aversion or repulsion, to empathy evidenced by feelings of affection, enchantment and interest in the animal. In this sense, herein we investigated the perception of university students about species belonging to different classes of wild vertebrates and the influence of social and educational factors on that.

**Methods:** Data were obtained through online forms answered by 700 university students from nine Brazilian states, 328 females and 372 males, aged between 18 and 65 years. The form had eight sentences to be answered in relation to 17 species of wild vertebrates. The agreement level for each of these sentences was to be indicated using a five-point Likert scale. The sentences were designed to assess aesthetic, risk, utilitarian, and preservation perceptions attributed to each species by students.

**Results:** We found that species perceived as useful by the students are generally also perceived as beautiful and as those that should be preserved. On the other hand, we found similarity between the species perceived as ugly and those that should not be preserved; and between the species perceived as harmful and those considered dangerous. Female and lower-income students more often agree that animals are harmful. We found that perceptions of danger in relation to animals were predominantly associated with younger respondents. However, this did not lead to less support for conservation among these students, as students of all age groups agree that species should be preserved. Our results show that students' knowledge area was an important predictor associated with empathetic and antipathetic perceptions. Environmental area students showed greater empathy in all analyzed categories (beauty, usefulness, harmlessness, and preservation) than non-environmental areas students. On the other hand, students from the area of the exact sciences showed greater dislike in all analyzed categories than students from other areas. We found a strong relationship between the areas "Environmental" and "Humanities, Languages and Arts" for the attitudinal factors associated with utility
and preservation, suggesting a similar empathetic worldview for students in these areas.

**Conclusions:** We found that the perception directed towards wild vertebrates varies according to the gender, age, income and study area of the students, in addition to the taxon considered. Finally, our results indicate that negative perceptions should be taken into account in environmental education efforts, educational policies and in planning fauna conservation plans which should incorporate the most diverse audiences, and not only encompass charismatic species but extend to animals that arouse great aversion from the part of people.

## INTRODUCTION

The existing relationships between humans and non-human animals have been altered in the course of the social transformations that have taken place throughout human history (*Alves, 2012*; *Alves & Souto, 2015*). Naturally, these interactions vary according to the animal taxa involved and the cultural context in which human populations are inserted (*Alves & Albuquerque, 2018*; *Becerra, Marinero & Borghi, 2022*; *Roldán-Clarà et al., 2021*; *van Vliet et al., 2022*).

The vertebrate group includes taxa that stand out for providing useful products to humans, as well as species that are targets of various conflicts (*Gore et al., 2006*; *Inskip & Zimmermann, 2009*; *Alves, 2012*; *Mascarenhas-Junior et al., 2021*). The wide range of perceptions with a greater degree of aversion or greater degree of empathy of people regarding different vertebrate species influences human attitudes towards fauna (*Alves, 2012*; *Alves et al., 2010*). This can be influenced by physical, behavioral or cognitive similarities with humans (*Miralles, Raymond & Lecointre, 2019*), with empathic responses appearing to be more important for taxa that are closely related to us (*Prokop et al., 2021*; *Prguda & Neumann, 2014*).

It is known, for example, that people generally have a great aversion to snakes and have negative attitudes towards these animals (*Alves et al., 2009*, *Alves, Gonçalves & Vieira, 2012*; *Silva et al., 2021*). Another example is sharks being stigmatized as devouring humans, a situation partly influenced by negative contributions from the media, particularly television and cinema (*Neves, McGinnis & Giger, 2022*; *Ostrovski et al., 2021*). These circumstances make the conservation of the group have a very low popular appeal, unlike what happens with other marine animals such as turtles, dolphins and manatees (*Thompson & Rog, 2019*; *Lessa et al., 1999*). Thus, while some species are annually culled in numbers that put their populations at risk as a way to retaliate against human-wildlife conflicts, others are used as flagship species in conservation programs (*Sabino & Prado, 2006*; *Nogueira & Alves, 2016*; *Lima-Santos, Costa & Barros Molina, 2020*).

Formal education may have an effect on people's attitudes towards animals, indicating that a higher education level is reflected in more positive attitudes adopted by the person

(*Pinheiro, Rodrigues & Borges-Nojosa, 2016*; *Onyishi et al., 2021*; *Oliveira et al., 2019a*, *2019b*; *2020*). In this sense, *Ceríaco (2012)* indicated that people with higher education levels have fewer misconceptions about herpetofauna linked to folkloric aspects that can generate aversion, because education brings explanations from the real world, reducing myths that can be harmful to species. Gender also may influences the perception and attitudes toward wildlife (*Zank, Hanazaki & de Melo, 2021*; *Sousa, Silva & Ramos, 2021*), since in most societies there is a strong division in social roles between men and women, which can alter the construction of knowledge through different lived experiences (*Silva et al., 2016*; *Torres-Avilez et al., 2016*).

In this study, we investigated the perception of university students from Northeastern Brazil about species belonging to different classes of wild vertebrates and the influence of social and educational factors on the preservation of these animals. More specifically, we verified whether: (1) students' perceptions vary according to the animal taxon (fish, amphibians, reptiles, birds and mammals); (2) the perception changes according to the social profile of the interviewees (gender, age and socioeconomic profile); and (3) university students from the Environmental area have greater empathy towards wild vertebrates compared to students from Non-Environmental areas (Exact sciences, Humanities, Arts and Health).

## MATERIALS AND METHODS

### Study area
The data for this study were obtained through sharing online forms among students from higher education institutions in northeastern Brazilian states. This region comprises an area of 1.5 million km$^2$ and extends from about 02°54′ to 17°21′S and from 35° to 46°30′W, including nine states. It has a population of around 57 million inhabitants, representing approximately 27% of Brazil's population (*IBGE, 2021*).

### Participants
The form was answered by students from five knowledge areas: (1) Environmental ($N = 285$); (2) Linguistics, Languages and Arts ($N = 31$); (3) Exact and Technological Sciences ($N = 163$); (4) Health ($N = 107$); and (5) Applied Social Sciences and Humanities ($N = 114$). This categorization was based on the classification of the Capes 2017 Table (the most recent version available online: http://www.capes.gov.br/avaliacao/instrumentos-de-apoio/tabela-de-areas-do-conhecimento-avaliacao). The study obtained a favorable opinion from the Research Ethics Committee of the Health Sciences Center of the Federal University of Paraíba—CEP/CCS (Prot. No. 095/16. CAAE: 54452015.5.0000.5188).

Participants from each knowledge area were selected from web pages of undergraduate courses in the social media Facebook., for example "*Biologia* 2016", "*Engenharia Civil* 2016.2", "*Enfermagem* 2015", "*Letras* 2016". The Facebook pages were select using the name of undergraduate courses and universities located in nine states of Northeast Brazil. Once the pages were found, a request was sent to the group administrators and after acceptance into the groups, all members were invited to participate in the research. Data were collected after receiving written consent from the students, who were previously

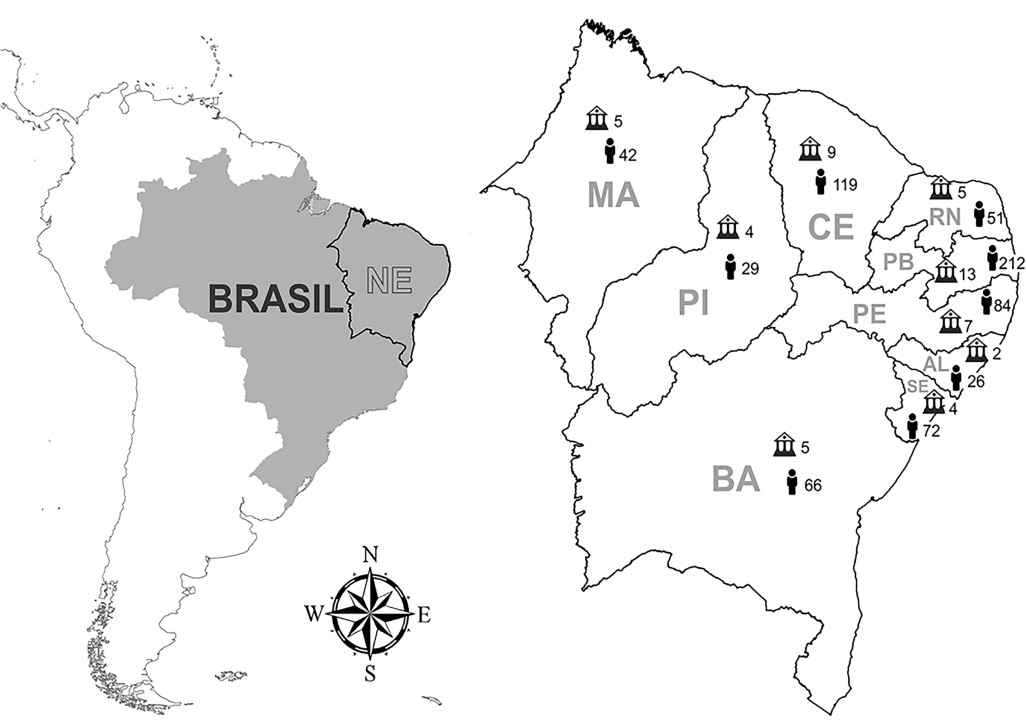

**Figure 1 States of location of respondents, number of institutions included in the research by state (N = 49).** Number of students interviewed by state (N = 700).

informed about the research objectives. Responses were obtained from 700 undergraduate students belonging to 61 courses from 49 institutions (18 Federal, 11 state and 20 private institutions) distributed in all of the Northeast Brazilian states (Fig. 1). Of the total number of respondents, 328 were women (46.9%) and 372 were men (53.1%), aged between 18 and 65 years. The questionnaires were applied from April 2016 to January 2017.

## Data collection

Students' perceptions of wild vertebrates were assessed through the agreement level to a series of sentences that indicate greater or lesser empathy and antipathy towards each species. The sentences were related to 17 animal species, including four mammals (dolphin, bat, jaguar and armadillo), four birds (vulture, owl, parrot and heron), four reptiles (snake, lizard, turtle and alligator), four fish (shark, ray, tilapia and piranha) and one amphibian (frog). The animals were chosen considering their potential to arouse different levels of empathy or aversion in students, in addition to being commonly found in urban and/or rural areas of the study region. Among the chosen animals, we sought to select some species considered as "charismatic", some others with utilitarian values for humans and others that are "target of conflicts" or arouse aversion by being historically stigmatized.

Four categories of perception were considered, each one consisting of two antagonistic sentences: (1) Aesthetic perception, measured by the sentences "*the animal is beautiful*" and "*the animal is ugly*"; (2) Risk perception, measured by the sentences "*the animal is*

*dangerous*" and "*the animal is harmless*"; (3) Utilitarian perception, measured by the sentences "*the animal is useful*" and "*the animal is harmful*"; and (4) Ecological perception measured by the sentences "*the animal must be preserved*" and "*the animal must not be preserved*". The empathy and aversion levels were measured according to the agreement level with each sentence, with an agreement degree ranging from 1 to 5 on the Likert scale (Strongly Disagree, Slightly Disagree, Neither Agree nor Disagree, Slightly Agree and Strongly Agree). We attributed the perception of empathy to agreement with the empathetic sentences and disagreement with the antipathetic sentences. We attributed the perception of aversion to agreement with antipathetic sentences and disagreement with empathetic sentences.

## Data analysis

Reliability analyzes of the scales used were performed considering Cronbach's alpha as an indicator of internal consistency (*Byrne, 2001*). The reliability test was applied to the four scale factors and to all factors together.

Next, we performed multiple factor analysis (MFA) (*Husson et al., 2018*) to check for similarity between the most frequent species in each perception category. MFA takes into account the fact that the data is structured into groups (herein, different species) to balance the importance of each group in the analysis. Some perception categories are then associated when the same agreement level with certain attitudes is answered for the same set of species.

The Mann-Whitney U test was used to test the variation in the perception of students of different genders in relation to each taxonomic group evaluated. To examine the effects of student gender, age, income and knowledge area on the level of agreement with each sentence we performed a set of cumulative link mixed model (CLMM). We used CLMM because the data of the level of agreement are ordinal, ranging from 1 (completely desagree) to 5 (completely agree) (*Braga-Pereira et al., 2022*). We performed five different models associated with each perception category. We considered (i) the perception category as a response variable; (ii) student gender, age, income and knowledge area as fixed effects predictor variables (There was no collinearity ($p > 0.05$) among predictor variables); and (iii) student code as a random factor. To verify whether our models were, in principle, suitable or not we used residual checks. We used the Akaike information criterion to select models when $\Delta AIC$ values >6 ($\Delta AIC$ obtained from the difference between a null and complete model AIC values; *Harrison et al., 2018*; *Richards, 2008*). All analyses were performed in R ver. 3.5.3 (*R Development Core Team, 2019*) CLMM were based on the ordinal package (*Christensen, 2019*) and MFAs were based on the FactoMineR (*Husson et al., 2018*).

# RESULTS

## Similarities between species in different perception categories

We found that species for those students agree are useful are generally also perceived as beautiful and as those that should be preserved. On the other hand, we found similarity between the species perceived as ugly and those that should not be preserved; as well as

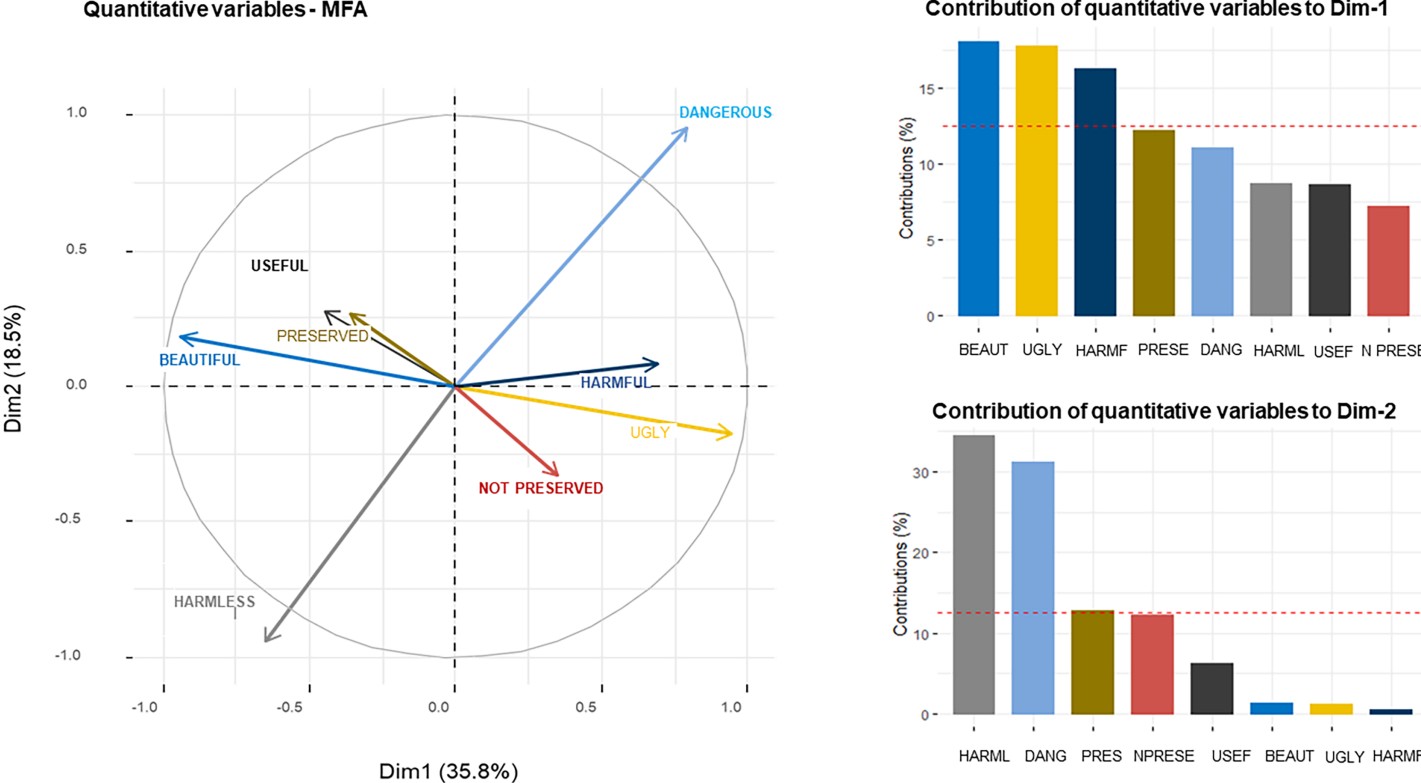

**Figure 2 Compositional similarity of sentences of perceptions of empathy and antipathy obtained from (A) of the MFA.** Contributions of each perception to (B) the first (Dim1) and (C) the second dimension (Dim2). The red dotted line indicates the percentage that would be obtained if all factors contributed equally to the overall variance.

between species perceived as harmful and those considered dangerous (Fig. 2). The first dimension (Dim$_1$) of the MFA accounted for 35.8% of the variability in attitude categories across different species (Fig. 2A). The items beautiful, ugly and harmful contributed to 21%, 20% and 18% of the variance explained by Dim$_1$, respectively (Fig. 2B). The second dimension (Dim$_2$) of the MFA explained 18.5% of the variance with harmless and dangerous explaining most of the variance (34% and 31%, respectively) (Fig. 2C). We found a high frequency of responses regarding the agreement that the animal is useful and that it should be preserved for all species (Figs. 3A and 3G). In addition, most students disagreed that the species should not be preserved (Fig. 3H). Larger animals, such as jaguars, dolphins, turtles and sharks comprise the animals most frequently perceived as beautiful (Fig. 3C); while piranhas, vultures, bats and frogs are generally perceived as ugly (Fig. 3D). Species perceived as harmless are dispersed in all categories (Fig. 3E) and therefore the axis related to the harmless category did not appear close to the axes associated with other categories in the MFA analysis (Fig. 2). Snake, piranha and alligator make up the species most perceived as both dangerous and harmful (Figs. 3B and 3F).
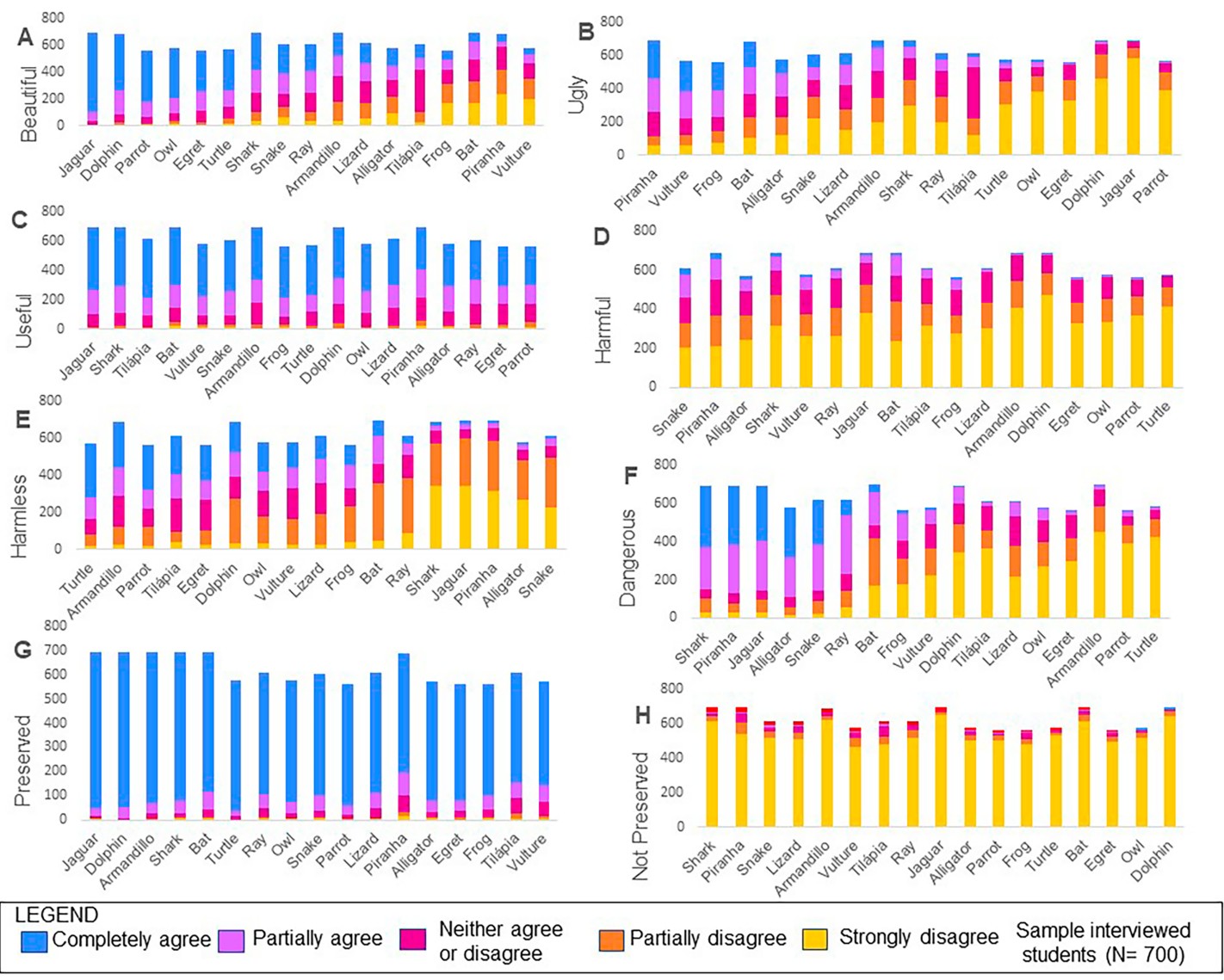

**Figure 3 Number of responses for each level of agreement associated with the perception categories Aesthetics (A and B), Utilitarian (C and D), Risk (E and F) and Preservation (G and H) among wild vertebrates.**

## Influence of socioeconomic factors on aesthetic perceptions of wild vertebrates

We found a trend that women agree more often than men that animals are "beautiful" and less often that they are ugly, but this difference between genders was not significant (Table 1). When dividing the species into taxonomic groups, we only observed differences in perceptions between genus for mammals. In this case, we found that women more often agreed that the mammal species presented are beautiful, which denotes empathy (W = 21,579; $p$ = 0.01). On the other hand, men more often agreed that these species are ugly, indicating dislike (W = 16,158; $p$ = 0.01). There was no statistically significant variation in relation to the aesthetic category for the other taxa. The variation in the age group and income of the interviewees did not interfere in the aesthetic preference

**Table 1 Ordinal models designed to verify the effect of age, gender, area of study and income of respondents in relation to Aesthetics.**

| Response variable | Predictor variables | Estimate | Std. error | z value | Pr(>|z|) | | AIC | AIC null model | ΔAIC |
|---|---|---|---|---|---|---|---|---|---|
| Beautiful | Age | 0.002861 | 0.007376 | 0.388 | 0.69812 | | 30,630.47 | 30,744 | 113.53 |
| | Male: Female | −0.00453 | 0.08315 | −0.054 | 0.95659 | | | | |
| | A: CET | −1.06814 | 0.106636 | −10.017 | <2.00E−16 | *** | | | |
| | CSAH: A | −0.78108 | 0.117514 | −6.647 | 3.00E−11 | *** | | | |
| | LLA: A | −0.53802 | 0.200009 | −2.69 | 0.00715 | ** | | | |
| | S: A | −0.94677 | 0.120177 | −7.878 | 3.32E−15 | *** | | | |
| | Family income | 0.020939 | 0.0264 | 0.793 | 0.4277 | | | | |
| Ugly | Age | −0.00057 | 0.007224 | −0.079 | 0.93676 | | 30,467.14 | 30,586.18 | 119.04 |
| | Male: Female | 0.041412 | 0.081418 | 0.509 | 0.61101 | | | | |
| | CET: A | 1.009141 | 0.104326 | 9.673 | <2.00E−16 | *** | | | |
| | CSAH: A | 0.865941 | 0.115195 | 7.517 | 5.60E−14 | *** | | | |
| | LLA: A | 0.547248 | 0.19635 | 2.787 | 0.00532 | ** | | | |
| | S: A | 0.993913 | 0.117561 | 8.454 | <2.00E−16 | *** | | | |
| | Family income | −0.0359 | 0.025898 | −1.386 | 0.16572 | | | | |

**Note:**

Abbreviations: E, Environmental; CET, Exact and Technological Sciences; CSAH, Applied Social Sciences and Humanities; LLA, Letters, Linguistics and Arts; S, Health. Pr (>|z|) are listed as two-tailed p values that correspond to the z values following a normal distribution pattern. Significance levels were as follows: ns $p > 0.05$; $p \leq 0.05$; **$p \leq 0.01$; **$p \leq 0.001$.

of the animals ($p > 0.05$). Students from all areas when compared to students from environmental areas significantly perceive that the analyzed animals are uglier (Table 1).

### Effect of socioeconomic factors on risk perceptions

Age showed an effect on students' aversion, with younger students showing a greater tendency to fear, confirmed by the item "Dangerous" when compared to older students who attributed the perception that animals are more "harmless" with greater frequency, however this tendency was not significant. We also found no significant difference between income and gender regarding risk factors (Table 2). However, analysis by taxonomic group showed that male respondents were less likely to have aversions to reptiles than females. In turn, females more frequently attributed "Dangerous" to animals in this group ($W = 15,614$; $p = 0.01$), to the detriment of animals from the other taxonomic groups analyzed. Therefore, we did not obtain significant variation in relation to the other taxa. Regarding the knowledge area, students from all other areas perceive animals as more dangerous and less harmless when compared to students from the environmental area.

### Effect of socioeconomic factors on utilitarianism perceptions attributed to taxa

There was an influence of gender and student income on the harmful category, with women and lower-income students more often agreeing that animals are harmful. Regarding the knowledge area, the agreement was significantly higher when considering the usefulness of vertebrates and lower when considering their harmfulness between students of courses in the environmental area in relation to those in the areas of "Exact and

**Table 2 Ordinal models designed to verify the effect of age, gender, area of study and income of respondents in relation to Risk.**

| Response variable | Predictor variables | Estimate | Std. error | z value | Pr(>|z|) | | AIC | AIC null model | ΔAIC |
|---|---|---|---|---|---|---|---|---|---|
| Dangerous | Age | −0.0128 | 0.005499 | −2.328 | 0.0199 | * | 32,627.02 | 32,687.06 | 60.04 |
| | Male: Female | −0.07712 | 0.061621 | −1.251 | 0.2108 | | | | |
| | CET: A | 0.550612 | 0.07922 | 6.95 | 3.64E−12 | *** | | | |
| | CSAH: A | 0.415412 | 0.08732 | 4.757 | 1.96E−06 | *** | | | |
| | LLA: A | 0.380898 | 0.148025 | 2.573 | 0.0101 | * | | | |
| | S: A | 0.49943 | 0.08918 | 5.6 | 2.14E−08 | *** | | | |
| | Family income | 0.024774 | 0.019653 | 1.261 | 0.2075 | | | | |
| Harmless | Age | 0.017547 | 0.006197 | 2.832 | 0.00463 | ** | 32,810.42 | 32,828.69 | 18.27 |
| | Male: Female | 0.043145 | 0.069681 | 0.619 | 0.5358 | | | | |
| | CET: A | −0.38918 | 0.089677 | −4.34 | 1.43E−05 | *** | | | |
| | CSAH: A | −0.18455 | 0.098673 | −1.87 | 0.06144 | . | | | |
| | LLA: A | −0.35428 | 0.166569 | −2.127 | 0.03343 | * | | | |
| | S: A | −0.32648 | 0.100956 | −3.234 | 0.00122 | ** | | | |
| | Family income | −0.01403 | 0.022218 | −0.631 | 0.52771 | | | | |

**Note:**
Abbreviations: A, Environmental; CET, Exact and Technological Sciences; CSAH, Applied Social Sciences and Humanities; LLA, Letters, Linguistics and Arts; S, Health.
Pr (>|z|) are listed as two-tailed $p$ values that correspond to the z values following a normal distribution pattern. Significance levels were as follows: ns $p > 0.05$; $p \leq 0.05$; **$p \leq 0.01$; **$p \leq 0.001$.

**Table 3 Ordinal models designed to verify the effect of age, gender, area of study and income of respondents in relation to utilitarianism.**

| Response variable | Predictor variables | Estimate | Std. error | z value | Pr(>|z|) | | AIC | AIC null model | ΔAIC |
|---|---|---|---|---|---|---|---|---|---|
| Useful | Age | 0.007104 | 0.017166 | 0.414 | 0.679 | | 19,188.76 | 19,246.8 | 58.04 |
| | Male: Female | 0.186096 | 0.195761 | 0.951 | 0.3418 | | | | |
| | CET: A | −1.9796 | 0.249857 | −7.923 | 2.32E−15 | *** | | | |
| | CSAH: A | −1.19782 | 0.274287 | −4.367 | 1.26E−05 | *** | | | |
| | LLA: A | −0.78058 | 0.464509 | −1.68 | 0.0929 | . | | | |
| | S: A | −1.50345 | 0.282192 | −5.328 | 9.94E−08 | *** | | | |
| | Family income | 0.050179 | 0.061637 | 0.814 | 0.4156 | | | | |
| Harmful | Age | −0.00161 | 0.001336 | −1.206 | 0.227728 | | 22,645.07 | 22,679.13 | 34.06 |
| | Male: Female | −0.32636 | 0.001875 | −174.035 | <2.00E−16 | *** | | | |
| | CET: A | 1.00767 | 0.001875 | 537.439 | <2.00E−16 | *** | | | |
| | CSAH: A | 0.618616 | 0.181239 | 3.413 | 0.000642 | *** | | | |
| | LLA: A | 0.577028 | 0.337199 | 1.711 | 0.087038 | . | | | |
| | S: A | 1.038877 | 0.183769 | 5.653 | 1.58E−08 | *** | | | |
| | Family income | −0.19047 | 0.003324 | −57.301 | <2.00E−16 | *** | | | |

**Note:**
Abbreviations: A, Environmental; CET, Exact and Technological Sciences; CSAH, Applied Social Sciences and Humanities; LLA, Letters, Linguistics and Arts; S, Health.
Pr (>|z|) are listed as two-tailed $p$ values that correspond to the z values following a normal distribution pattern. Significance levels were as follows: ns $p > 0.05$; $p \leq 0.05$; **p $\leq 0.01$; **p $\leq 0.001$.

Technological Sciences", "Applied Social Sciences" and "Human Sciences". On the other hand, we did not find a significant difference between students in the "Languages, Linguistics and Arts" area and in the "Environmental" area (Table 3).

**Table 4 Ordinal models designed to verify the effect of age, gender, area of study and income of respondents in relation to preservation.**

| Response variable | Predictor variables | Estimate | Std. error | z value | Pr(>\|z\|) | | AIC | AIC null model | ΔAIC |
|---|---|---|---|---|---|---|---|---|---|
| Preserved | Age | −0.03492 | 0.01873 | −1.865 | 0.0622 | . | 9,492.07 | 9,502.07 | 10 |
| | Male: Female | 0.05516 | 0.21735 | 0.254 | 0.7996 | | | | |
| | CET: A | −1.32584 | 0.27351 | −4.847 | 1.25E−06 | *** | | | |
| | CSAH: A | −0.68263 | 0.30408 | −2.245 | 0.0248 | * | | | |
| | LLA: A | 0.68779 | 0.5541 | 1.241 | 0.2145 | | | | |
| | S: A | −0.768 | 0.31445 | −2.442 | 0.0146 | * | | | |
| | Family income | 0.06078 | 0.06865 | 0.885 | 0.376 | | | | |
| Not preserved | Age | 0.06046 | 0.01732 | 3.49 | 0.000482 | *** | 9,469.7 | 9,486.48 | 16.78 |
| | Male: Female | −0.08506 | 0.20434 | −0.416 | 0.677204 | | | | |
| | CET: A | 0.75854 | 0.26026 | 2.915 | 0.003562 | ** | | | |
| | CSAH: A | 0.74505 | 0.28557 | 2.609 | 0.009082 | ** | | | |
| | LLA: A | −0.68934 | 0.51888 | −1.329 | 0.184009 | | | | |
| | S: A | 0.83797 | 0.29274 | 2.863 | 0.004203 | ** | | | |
| | Family income | −0.1332 | 0.06521 | −2.043 | 0.041073 | * | | | |

**Note:**

Abbreviations: A, Environmental; CET, Exact and Technological Sciences; CSAH, Applied Social Sciences and Humanities; LLA, Letters, Linguistics and Arts; S, Health.
Pr (>\|z\|) are listed as two-tailed $p$ values that correspond to the z values following a normal distribution pattern. Significance levels were as follows: ns $p > 0.05$; $p \leq 0.05$;
**$p \leq 0.01$; **$p \leq 0.001$.

### Influence of socioeconomic factors on preservation perceptions

Our results showed that university students interviewed in general support the preservation of the evaluated organisms (median = 4.90; interquartile range = 4.70 to 5.0). We found that the agreement regarding the preservation of wild vertebrates is significantly higher among students of courses in the Environmental area in relation to those in the Exact and Technological Sciences area, followed by students of the Applied Social Sciences and Humanities area; and finally, by Health. On the other hand, we found no significant difference between students in the Languages, Linguistics and Arts area and in the Environmental area regarding the preservation of wild vertebrates.

We also observed that agreement of the "do not preserve" category was significantly lower among students from courses in the Environmental area in relation to those in the areas of Exact and Technological Sciences, Applied Social Sciences and Humanities. Older and lower-income students more often agree that species should not be preserved (Table 4).

### Cronbach's alpha for scale factors

The factor analysis results indicated that the eight items used in our scale were used to measure four factors: Aesthetics, Utility, Risk and Preservation. Cronbach's alpha coefficient for the Aesthetic Factor was (α = 0.94); we obtained (α = 0.79) for the Risk factor, (α = 0.81) for the Preservation factor, while we found a much lower value (α= 0.17) for the Utility factor. Cronbach's alpha coefficient for the entire instrument was (α = 0.73). This indicates that the questionnaire showed a good degree of reliability (*Nunnally, 1978*; *Prokop et al., 2010*; *Prokop, Özel & Usak, 2009*).

## DISCUSSION

We found that factors such as aesthetics, utilitarian potential and the risk associated with animals are strong influencers of perceptions and attitudes towards the preservation of wild vertebrates. These results reinforce previous studies carried out with students of different education levels from kindergarten to University (*Kidd & Kidd, 1990*; *Prokop, Fancovicova & Kubiatko, 2009*; *Prokop, Özel & Usak, 2009*; *Campos et al., 2012*; *Oliveira et al., 2020*; *Alves et al., 2014*; *Borgi & Cirulli, 2015*; *Kubiatko, 2012*).

It is known that the perception directed to animals varies according to taxa. Our results reinforce this trend, with a great variation in the perception of the students interviewed in relation to the different animals analyzed. Among the animals considered the most beautiful are large, charismatic and iconic mammals such as the jaguar and dolphin, and one of the most popular wild birds as pets in Brazil and in the world: the parrot (*Alves et al., 2013*; *Fernandes-Ferreira et al., 2012*; *Roldán-Clarà et al., 2014*). This is expected once large-sized mammals are among the most charismatic animals (*Albert, Luque & Courchamp, 2018*; *McGowan et al., 2020*), which are often the target of conservation campaigns (*Kontoleon & Swanson, 2003*; *Clucas, McHugh & Caro, 2008*; *Barney, Mintzes & Yen, 2005*; *Schlegel & Rupf, 2010*; *McGowan et al., 2020*).

In another perspective, some authors suggest that humans empathize with phylogenetically closer animals, for example those exhibiting physical, behavioral or cognitive similarities with humans (*Herzog & Burghardt, 1988*; *Miralles, Raymond & Lecointre, 2019*; *Prokop et al., 2021*). Therefore, mammals would also benefit in this aspect due the emotional perceptions we can feel for them which is largely related to the anthropomorphic projections that we put on mammals, such as attribution of human traits, emotions, or intentions to non-human entities (*Miralles, Raymond & Lecointre, 2019*). This imbalance is so marked that even scientific research on biodiversity or conservation efforts present a significant bias in favor of our societal inclinations for particular taxa (*Colléony et al., 2017*; *Troudet et al., 2017*).

On the other hand, among the taxa with the lowest scores in relation to aesthetics are animals such as piranha, vultures, frogs and bats, which are associated with risks to humans or negative beliefs. For example, bats are the target of legends and tales that encourage people to dislike these animals in Brazil (*Rego et al., 2015*), being considered uncharismatic for the general population and the potential ecological benefits of these animals are rarely disclosed.

It is worth mentioning that some specie's attributes, like morphology and coloration, are factors that influence on the society's perception and their attitudes towards fauna (*Prokop & Fančovičová, 2013*; *Fráncel et al., 2021*; *Castilla et al., 2020*). Aesthetically attractive species also received greater support for preservation and were more perceived as useful by respondents. Other studies (*Pinho et al., 2014*; *Gunnthorsdottir, 2001*) have also reported greater public support for species deemed most attractive. Our data also revealed that students of both genders valued the aesthetics of organisms, but with women considering mammals more beautiful and reptiles more dangerous than other taxa. For *Kaltenborn et al. (2006)*, gender exerts a significant influence on affinity levels towards animals. In this

sense, some works such as those by (*Kellert, 1989*; *Williams, Ericsson & Heberlein, 2002*; *Zinn & Pierce, 2002*) suggest that women have higher levels of support for species protection than men. However, it is noteworthy that this affinity may vary according to the taxon considered, as found herein.

The results obtained by *Czech, Krausman & Borkhataria (1998)* reinforce this situation, suggesting that mammals, birds and fish are part of a distinctly more positive social construction, and thus were identified as more "advantaged" than reptiles (excluding turtles, which is considered a charismatic species) (*Alves, et al., 2012a*, *2012b*, *2014*). Turtles are characterized by docility and are often the target of conservation campaigns, which reflects people's positive view of these chelonians (*Senko et al., 2011*; *Gamba et al., 2022*). As our results indicate, which point to the "turtle" with greater preservation appeal than other reptiles.

Like aesthetics, the usefulness of animals tends to be a factor which positively influences their perception by people. Species of practical utility in human life, whether for providing products used by people for nutrition or as part of their leisure activities tend to be more valued. In fact, the animal that obtained the highest score in the "useful" item was a fish (tilapia), widely consumed as food in Brazil. However, animals such as jaguars, vultures and frogs also achieved high scores, suggesting that students recognize the value of these animals in the environment. This may indicate that higher education students recognize the ecological role of animals in general, regardless of whether the animals in question are a source of products used directly by humans or not. Similar results were found by other authors, such as *Kellert & Berry (1980)*, *Bjerke & Ostdahl (2004)* and *Schlegel & Rupf (2010)*, who found coherence between higher education levels and positive perceptions of fauna.

If people has a greater empathy with useful animals, humans establish a conflict relationship with animals of potential risks to them, in turn, establishing more aversion with those species (*Onyishi et al., 2021*; *de Oliveira et al., 2019*; *Oliveira et al., 2020*; *Vergara-Ríos et al., 2021*). This is the case of snakes in Brazil, which, as some works point out, are the target of aversion and fear on the part of people generally due to the risk they pose to human lives and their domestic animals (*Alves, 2012*; *Fernandes-Ferreira et al., 2013*; *Mendonça, Vieira & Alves, 2014*). Additionally, snakes inspire many myths, proverbs and stories that are transmitted orally and that place these animals as beings associated with evil and that influence the way local people relate to these animals, generally provoking negative attitudes on the part of people (*Fernandes-Ferreira et al., 2013*; *Mendonça et al., 2012*; *Mendonça, Vieira & Alves, 2014*; *Lima-Santos, Costa & Barros Molina, 2020*). This explain why snakes were the animals which generally had the highest scores on the harmful item among the students interviewed. In addition to snakes, animals such as piranhas, alligators and sharks, which can be seen as a potential risk to humans, were recognized as more harmful.

Our data revealed that lower-income students more often agree that animals are harmful. This result can be associated with the fact that the economic impact generated by the attack of wild animals on crops and livestock is more significant for low-income people than for those with higher incomes. From another perspective, among the animals that

were shown to arouse higher perceptions of danger in the students interviewed, some of them such as sharks, jaguars, alligators and snakes are known to evoke fear because they are predominantly large, with physical characteristics that arouse threat or because they are venomous (*Staňková et al., 2021*; *Silva et al., 2021*). A greater perception of fear and harm by smaller vertebrates was associated with piranha, which may indicate a direct threat link to humans given that their attack can generate extensive tissue loss and bleeding (*Haddad & Sazima, 2003*). Piranhas are quite small compared to sharks (for example), but both fish stand out in our study as the species with the highest fear perception scores. Both species are carnivorous, with sharp teeth and often referred to as dangerous animals by the media, thereby constituting characteristics which reinforce the perception of fear in humans. It should be noted that other animals such as bats and snakes, for example, are also associated with negative aspects by the media, enhancing society's negative perception of them.

Negative perceptions directed towards animals imply less support for their preservation, and, as we have pointed out in our study, may result from social factors such as gender, income, age, superstitions and myths, formal education and education area. Reinforcing previous research, our results suggest that human beliefs which negatively impact animals are randomly pervasive in the population, regardless of age, culture, gender (*Mintzes, Wandersee & Novak, 1998*) or specialized study area (*Prokop, Fancovicova & Kubiatko, 2009*). The increase in knowledge about the ecological role of less attractive animals from the human perspective could make it possible to improve their respective images, reinforcing the need to create environmental policies that include less aesthetically attractive organisms, as well as flagship species.

Corroborating our results, previous studies have found that younger people with higher education are often more associated with positive perceptions of wildlife (*Dressel, Sandström & Ericsson, 2015*; *Kellert & Berry, 1980*; *Smith, Nielsen & Hellgren, 2014*; *Vaske, Jacobs & Sijtsma, 2011*). However, this has not led to less support for conservation among older students, as students of all age groups agree with high frequency that species should be preserved.

Our results also showed that the knowledge area of university students has an influence on perceptions of vertebrate preservation. In this sense, perceptions of preservation are more favorable among students of courses with content associated with nature and with greater contact with educational-environmental activities. Several studies point out that having contact with nature and developing an emotional bond with natural elements are determining factors for a preservationist perception (*Collado, Staats & Corraliza, 2013*; *Collado et al., 2015*; *Duerden & Witt, 2010*). Thus, attitudes that support biodiversity will be consolidated through concrete experiences with nature (*Turpie, 2003*). In this context, since our results showed similarity in the preservation perception between students of "Languages, Linguistics and Arts" and the "Environmental" areas, it is assumed that a greater appreciation for biodiversity may also result from intrinsic motivations of the student's own subject, as they seek to study something for which they are already close to. On the other hand, direct contact with the urbanized environment and the appropriation of technologies, whether in the home or work environment, may imply less contact

between people and nature, both from an affective and preservation point of view (*Zhang, Goodale & Chen, 2014*). This situation may explain the greater antipathy towards the conservation of fauna observed among university students of the courses in the "Exact and Technological Sciences" area, therefore configuring a different connection link with the fauna than that established among the students of the "Environmental" and "Languages, Linguistics and Arts" areas.

A practical reflection of this result is that most of the infrastructure projects led by professionals who are not of the environmental area use minimal strategies to mitigate the impact of the projects on the fauna (*Lauxen, 2012*). Evidently, during the environmental licensing process, environmental area professionals must be involved, but the mitigation process would become more efficient if professionals who are leading the project (most often civil engineers) were also concerned with biodiversity conservation. For example, the estimates of roadkilled animals are 55 reptiles/km/year (*Gonçalves et al., 2017*) and 0.6 mammals/km/year (*Abra et al., 2021*). Although road mitigation strategies have the potential to reduce road-kill by up to 86%, few or no effective mitigation measures have been taken by the professionals responsible for these projects (*Rytwinski et al., 2016*).

## CONCLUSIONS

Our results indicate that the perception directed towards wild vertebrates varies according to the student's gender, age, and study area, in addition to the taxon considered, implying more or less favorable perceptions of animal preservation. Animals with utilitarian value and components of the so-called charismatic megafauna tend to have more preservation appeal, to the detriment of animals that cause aversion and are the target of constant conflicts with humans, such as snakes and bats, which confirms our hypothesis that there is a variation of perception directed to vertebrates according to the analyzed taxon. It is therefore evident that this whole scenario, which influences negative perceptions, must be taken into account in elaborating environmental education and fauna conservation projects, which must incorporate the most diverse audiences, and not only encompass charismatic species but also extend to animals that arouse great aversion on the part of people. Finally, we suggest an increase for a more multidisciplinary curriculum of professionals in the areas of engineering that reinforces the importance of biodiversity conservation, including for the economic and social development of the country.

## ACKNOWLEDGEMENTS

We are especially grateful to the university students from different institutions who supported us in collecting data and shared their knowledge about wild vertebrates with us. Without them, this work would not be possible. We thank LECOPSI—Laboratory of Behavioral Ecology and Psychobiology—UFPB for the support with access to the use of the laboratory's online software during the construction and execution of the study and for the guidance in the data tabulation. We are very grateful to the reviewers Pavol Prokop and Kiran Liversage for their effort to improve our manuscript.

### Funding
The authors received no funding for this work.

### Competing Interests
The authors declare that they have no competing interests.

### Author Contributions
- Heliene Mota Pereira conceived and designed the experiments, performed the experiments, analyzed the data, prepared figures and/or tables, authored or reviewed drafts of the article, and approved the final draft.
- Franciany Braga-Pereira analyzed the data, prepared figures and/or tables, authored or reviewed drafts of the article, and approved the final draft.
- Luane Maria Melo Azeredo performed the experiments, analyzed the data, authored or reviewed drafts of the article, and approved the final draft.
- Luiz Carlos Serramo Lopez conceived and designed the experiments, authored or reviewed drafts of the article, and approved the final draft.
- Rômulo Romeu Nóbrega Alves conceived and designed the experiments, authored or reviewed drafts of the article, and approved the final draft.

### Human Ethics
The following information was supplied relating to ethical approvals (*i.e.*, approving body and any reference numbers):

Ethics committee of my university

### Data Availability
The raw data are available in the Supplemental File.

### Supplemental Information
Supplemental information for this article can be found online at http://dx.doi.org/10.7717/peerj.14553#supplemental-information.

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
