# Peer review of "Assessing factors influencing students’ perceptions towards animal species conservation"

_PeerJ, doi:10.7717/peerj.14553_

## Round 0.1 · original submission · Minor Revisions

The research results presented by you are complete, well structured and relevant. The evaluation of the factors affecting the student's perception of the preservation of animal species indicates their diversity. I consider the use of the student survey method and the wide geography of the respondents to be effective.

The obtained research results are a valuable contribution to the understanding of students' perception of the problem of biodiversity conservation. The authors' materials correspond to the issues of the PeerJ journal. After detailed processing of the article and analysis of reviewers' feedback, I send the article to the authors for minor revision. Please follow the reviewers' recommendations.

Reviewer 1 has suggested that you cite specific references. You are welcome to add them if you believe they are relevant. However, you are not required to include these citations, and if you do not include them, this will not influence my decision

# ·

Basic reporting

This manuscript investigates human attitudes toward wild vertebrates with respect to gender, educational level, age etc. The sample sizes are high and statistical analyses sound appropriate. Results showed that charismatic megafauna received greater preservation appeal, whole dangerous animals cause rather aversion. Human willingness to protect animals was influenced by physical attractiveness of particular taxa.

Experimental design

The authors used a survey with large samples with well balanced sexes and various age groups. Data were collected online. Research questions were clear and easy to follow.

Validity of the findings

Results largely support previous findings regarding influences of animal taxa being considered (charismatic mammals scoring more than unpopular animals) and some gender differences etc. I very like explanations of the influence of socioeconomic factors on perception of animals (poorer people are more vulnerable to animal harm, thus perceive them less positively).

Additional comments

I have minor comments regarding missing references:

L. 72: I recommend to add further supportive refs:
Miralles, A., Raymond, M., and Lecointre, G. (2019). Empathy and compassion
toward other species decrease with evolutionary divergence time. Sci. Rep.
9:19555.
Prokop, P., Zvaríková, M., Zvarík, M., Pazda, A., & Fedor, P. (2021): The Effect of Animal Bipedal Posture on Perceived Cuteness, Fear, and Willingness to Protect Them. Frontiers in Ecology and Evolution, 9: 681241.

L. 82-83 – please consider also classic work showing this relationship in:
Kellert, S. R. (1993). Values and perceptions of invertebrates. Conservation Biology, 7(4), 845-855.

L. 84-87 – please consider also Onyishi, I. E., Nwonyi, S. K., Pazda, A., & Prokop, P. (2021): Attitudes and behaviour toward snakes on the part of Igbo people in southeastern Nigeria. Science of the Total Environment, 763, 143045. where education was associated with tolerance of snakes

Please creater one para in the Intro where you wil introduce your expectations regarding gender differences, because gender appears to be important predictor in the results.

Several references coted in the text are missing in the list of references. See e.g.:
Kidd & Kidd 1990; Prokop et al. 2009a; Prokop et al. 2009, Campos et al. 2012, Borgi and Cirulli 2015; Kubiatko 2012.

Please double-check references in the text and at the end of the MS.

L. 266-269 please consider also
Albert, C., Luque, G. M., & Courchamp, F. (2018). The twenty most charismatic species. PloS one, 13(7), e0199149.

L. 270: Please consider also:
Clucas, B., McHugh, K., & Caro, T. (2008). Flagship species on covers of US conservation and nature magazines. Biodiversity and Conservation, 17(6), 1517-1528.
Kontoleon, A.; Swanson, T. The willingness to pay for property rights for the giant panda: Can a charismatic species be an instrument for nature conservation? Land Econ. 2002, 79, 483–499.

L. 271: are also favored in relation
to society's perception of fauna - please consider also: Fančovičová, J., Prokop, P., Repáková, R., & Medina-Jerez, W. (2021). Factors Influencing the Sponsoring of Animals in Slovak Zoos. Animals, 12(1), 21.

L. 274 and 276 – again also please consider Prokop, P., Zvaríková, M., Zvarík, M., Pazda, A., & Fedor, P. (2021): The Effect of Animal Bipedal Posture on Perceived Cuteness, Fear, and Willingness to Protect Them. Frontiers in Ecology and Evolution, 9: 681241.

L. 283 – please also consider Prokop, P., & Fančovičová, J. (2013). Does colour matter? The influence of animal warning coloration on human emotions and willingness to protect them. Animal Conservation, 16(4), 458-466.

L. 296-8: For willingness to protect sea turtles, please consider: Senko, J., Schneller, A. J., Solis, J., Ollervides, F., & Nichols, W. J. (2011). People helping turtles, turtles helping people: understanding resident attitudes towards sea turtle conservation and opportunities for enhanced community participation in Bahia Magdalena, Mexico. Ocean & Coastal Management, 54(2), 148-157.

L. 313-318 – again please consider Onyishi et al. 2021 suggested above

L. 341-2: Mintzes & Wandersee 1998, Prokop & Kubiatko 2009 – missing in list of references, please improve

L. 343 - attractive animals in human vision could – perhaps from human point of view would be better than human vision

·

Basic reporting

There are just two comments I have here:

Title - correct the use of the apostrophe, i.e. make the title: "Assessing factors influencing students' perceptions towards animal species conservation"

line 182-183 - change to "Fig. 2a"

Experimental design

line 118 - here it states that participants were randomly selected - please could you provide some details about the specific randomization process for making sure the participants were randomly selected.

line 178-180 - at this point in reading the manuscript I starting thinking how it may have been more useful to have some more scientific method for choosing the species to include. For example, if the study was meant to be relevant to species commonly found in urban and/or rural areas of the study region, then a comprehensive list of these species could have been compiled, and a subset of species randomly selected from the list. This would prevent potential bias on the part of the authors while choosing the different types of species, which might have been chosen to fit into certain preconceived trends. For example, it would be difficult to choose a species like vultures without knowing full well that people will consider it ugly and probably not a priority for preservation. If species were randomly chosen, then some unexpected species might have been included; one example I thought of was wild pigs, which are probably considered ugly but useful for hunting and selling economically. Adding a few others like this may have been enough to alter the overall association found between species being ugly and not preserved (Fig. 2a). I ask the authors to please at least add some text describing why their choice of species for the experiment is not biased, and how it may do a good job of representing vertebrates in general from urban and/or rural areas of the study region.

Validity of the findings

no comment

Additional comments

line 40 - from the title it seemed that all the participants were students, but here it seems to the reader that some must not have been students, please clarify.

line 189 - is "macaws" meant to be included here? This is not an animal included specifically in the data, and I'm not sure if relative to the others on this list they could also be described as "larger animals".

---

## Round 0.2 · accepted · Accept

The finished materials of the article reveal the essence of the task set before the authors, who, with the help of a complex of current methods, were able to assess the factors affecting the students' perception of the preservation of animal species. The materials meet the requirements of PeerJ and are characterized by high scientific value.